# Microsatellite Status and IκBα Expression Levels Predict Sensitivity to Pharmaceutical Curcumin in Colorectal Cancer Cells

**DOI:** 10.3390/cancers14041032

**Published:** 2022-02-17

**Authors:** Lili Lu, Randy Przybylla, Yuru Shang, Meng Dai, Mathias Krohn, Oliver Holger Krämer, Christina Susanne Mullins, Michael Linnebacher

**Affiliations:** 1Department of General Surgery, Molecular Oncology and Immunotherapy, Rostock University Medical Center, 18057 Rostock, Germany; lili.lu@med.uni-rostock.de (L.L.); randy.przybylla@med.uni-rostock.de (R.P.); yuru.shang@med.uni-rostock.de (Y.S.); meng.dai@med.uni-rostock.de (M.D.); mathias.krohn@med.uni-rostock.de (M.K.); christina.mullins@med.uni-rostock.de (C.S.M.); 2Institute of Toxicology, Mainz University Medical Center, 55131 Mainz, Germany; okraemer@uni-mainz.de

**Keywords:** IκBα, NF-κB p65, microsatellite stability, microsatellite instability, curcumin, colorectal cancer, biomarker, sensitivity, IL-8, CXCL-1

## Abstract

**Simple Summary:**

The global burden of colorectal cancer is high. Chemotherapy has been the backbone of colorectal cancer therapy for decades. Toxic side effects and frequently occurring drug resistances remain challenging problems. Therefore, exploring natural compounds with low or even no toxicity holds great potential. However, natural curcumin is poorly absorbed, limiting its clinical use. Therefore, our focus was to screen different molecular types of colorectal cancer to find the ones with the highest sensitivity to curcumin. We observed very individual responses to curcumin for various colorectal cancer cell lines. Most curcumin-sensitive cell lines were of the microsatellite-stable molecular type, and expressed high baseline levels of the IκBα protein. Contrarily, curcumin-resistant lines were mainly microsatellite instable, with low baseline IκBα levels. Considering all of the data obtained, we conclude that patients with microsatellite-stable tumors and high baseline IκBα protein expression would benefit from treatment with novel curcumin formulations and derivatives.

**Abstract:**

Clinical utilization of curcumin in colorectal cancer (CRC) was revived as a result of the development of novel curcumin formulations with improved bioavailability. Additionally, identification of biomarkers for curcumin sensitivity would also promote successful clinical applications. Here, we wanted to identify such biomarkers in order to establish a predictive model for curcumin sensitivity. Thirty-two low-passage CRC cell lines with specified tumor characteristics were included. Curcumin suppressed cell proliferation, yet sensitivity levels were distinct. Most curcumin-sensitive CRC cell lines were microsatellite stable and expressed high levels of IκBα. The predictive capacity of this biomarker combination possessed a statistical significance of 72% probability to distinguish correctly between curcumin-sensitive and -resistant CRC cell lines. Detailed functional analyses were performed with three sensitive and three resistant CRC cell lines. As curcumin’s mode of action, inhibition of NF-κB p65 activation via IκBα was identified. In consequence, we hypothesize that novel curcumin formulations—either alone or, more likely, in combination with standard therapeutics—can be expected to prove clinically beneficial for CRC patients with high IκBα expression levels.

## 1. Introduction

In the database “Global Cancer Observatory”, the recent statistics for 2020 revealed that 1,931,590 individuals for both genders and all ages combined were diagnosed with colorectal cancer (CRC), thus accounting for approximately 10.0% of newly diagnosed cancer cases. More than 935,000 (9.4%) deaths of CRC patients per year make it the second most common cancer-related cause of death [1,2]. The highest CRC incidences and mortality rates mainly occurred in Europe [3]. The prognosis of CRC patients has progressively improved, especially due to early detection as well as advances in therapeutic strategies, including the most recent breakthrough in immunotherapy.

However, chemotherapy is still the backbone of clinical CRC management. Toxic side effects and drug resistances remain very challenging problems, thus limiting the efficacy of therapy and largely affecting patients’ quality of life [4,5,6]; therefore, even more effective treatments are required. This has also triggered the scientific exploration of natural compounds with alleged anticancer properties, which hold the promise of fewer side effects and lower toxicity. 

Curcumin, a natural yellow coloring agent, is a phytochemical derived from *Curcuma longa* (turmeric), which belongs to the ginger family [7,8]; it has repeatedly been reported as an agent affecting CRC in vitro. Curcumin inhibits growth, accelerates apoptosis, induces cell cycle arrest in the G2/M phase, and suppresses cell migration and invasion [9], and has been found to act on various signaling pathways and molecular targets, including enzymes (e.g., COX-2), proteases, transcription factors, Bcl-2 family members, death receptors, and reactive oxygen species [9,10].

In fact, most of the anti-inflammatory and antitumor properties of curcumin are considered to be mediated via blocking of the NF-κB pathway’s activation. This pathway has been shown to be upregulated in CRC, and is a major cause of drug resistance [11,12]. Zhou et al. [13] and Kunnumakkara et al. [14] found that curcumin exhibits antitumor and antimetastatic effects by directly inhibiting NF-κB p65 activation and impairing NF-κB p65-regulated gene expression, including *cyclin D1*, *Bcl-2*, *Bcl-xL*, *COX-2*, and *MMP-9*.

However, Wong et al. [15] reported that realistically achievable curcumin serum concentrations are only in the nanomolar range, due to poor absorption and low bioavailability. Presently, this weakness clearly limits curcumin’s clinical potential. 

However, novel drug formulations—such as Meriva^®^ and liposomal complexes, which are currently under investigation—promise the possibility of a “revival” of this very old natural compound for oncological purposes [12,16,17]. Identifying tumor- and patient-specific factors allowing the definition of specific CRC subsets with the highest sensitivity to curcumin would be another way to improve successful clinical application. 

As various CRC cell lines differ in their response to treatments due to genomic and phenotypic alterations and molecular mechanisms [18,19,20,21,22], we hypothesized that curcumin sensitivity might also be linked to specific molecular features such as mutations or activated molecular signaling pathways.

The goal of the present study was to (1) assess the sensitivity of CRC cell lines with defined molecular alterations to curcumin, (2) identify the mode of action by mechanistically analyzing sensitive CRC cell lines, (3) identify biomarkers associated with curcumin sensitivity, and (4) predict the sensitivity of additional CRC cell lines to curcumin by establishing a predictive model using the previously identified biomarkers.

## 2. Results

A total of 32 low-passage patient-derived cell lines, established from 30 colon cancer (CC) and 2 rectal cancer (RC) patients, were used in the present study [23]. CRC patients had a mean age of 66 years, with an almost equal distribution between males and females. All UICC stages were represented, and were distributed as follows: 40.63% stages I–II and 56.25% stages III–IV; one case (HHC6548) was unclear. After molecular classification [23], around 44% of the study population was categorized as sporadic standard (spStd), 25% as sporadic high-degree microsatellite instable (spMSI-H), 16% as Lynch-associated, 13% as highly CpG island methylated (CIMP-H), and 3% as neuroendocrine (Table 1).

### 2.1. Sensitivities to Curcumin and to Classical Chemotherapeutics Are Not Associated

All CRC cell lines were treated with increasing concentrations of curcumin, 5-FU, irinotecan, and oxaliplatin. The IC_50_ of curcumin ranged from 6.69 μM to 18.49 μM. Values of 0.17 μM to 80.95 μM for 5-FU, 0.04 μM to 12.80 μM for irinotecan, and <0.01 μM to 8.72 μM for oxaliplatin were observed [24,25,26,27,28] (Table 2). In general, the mean IC_50_ values of irinotecan and oxaliplatin were significantly lower than that of curcumin in CRC cells, with all *p*-values < 0.001. Using Spearman’s correlation analysis, no significant correlation between curcumin and any of the standard drugs was identified; the correlation coefficients were −0.12 for 5-FU (*p* = 0.545) (Figure 1A), −0.15 for irinotecan (*p* = 0.446) (Figure 1B), and −0.32 for oxaliplatin (*p* = 0.089) (Figure 1C); the latter might be indicative of a trend towards an inverse correlation.

### 2.2. Curcumin-Sensitive CRC Cell Lines Expressed Higher IκBα Levels

As shown in Figure 2A, a minority of CRC cell lines already responded at low concentrations, whereas 20% of the CRC cell lines were resistant even at the highest curcumin concentrations applied (Figure 2A). 

Considering the role of NF-κB in cancer progression, and the established effects of curcumin on the NF-κB pathway, we next measured baseline IκBα levels in all 32 CRC cell lines via flow cytometry. The IκBα expression levels ranged from 3.20% to 56.96% (Figure 2B). The level of baseline IκBα expression was inversely correlated with the curcumin IC_50_ concentrations. The correlation coefficient was −0.72 and *p* was < 0.001 (Figure 2C). This indicates that CRC cells with high baseline IκBα levels are more sensitive to curcumin than cells with low IκBα expression. 

### 2.3. Curcumin Induced Cell Death in Curcumin-Sensitive CRC Cell Lines

In order to mechanistically examine the different curcumin responses in sensitive and resistant CRC cell lines, three cell lines from each group were selected. The sensitive cell lines were HROC18, HROC69, and HROC357, and the resistant cell lines were HROC24, HROC113, and HROC285. All six cell lines were treated using 0 µM (control), 10 μM, 15 μM, and 20 μM curcumin for 72 h. As expected, cell death in HROC24, HROC113, and HROC285 was not significantly affected, even at the highest curcumin concentration of 20 μM. Conversely, the cell death rate was significantly higher in all three curcumin-sensitive cell lines compared to the control group (*p* < 0.05), and even an obvious dose-dependent response to curcumin was observed (Figure 3A). In HROC357, the strong curcumin response started at 10 μM curcumin (*p* < 0.05). Cell death increased to 80% dead cells in the HROC357 and HROC69 cell lines when treated with 20 μM curcumin (*p* < 0.05) (Figure 3A). 

### 2.4. Curcumin Suppressed Migration in Curcumin-Sensitive CRC Cell Lines

Confluent curcumin-sensitive (HROC18 and HROC357) and -resistant (HROC24 and HROC87) cells were treated with 0 µM (control) and 10 μM curcumin overnight, scratched, and the wounds’ diameters were subsequently measured daily for up to 5 days. Migration was not affected by curcumin in HROC24 and HROC87; however, in HROC18 and HROC357 cells, the cells’ ability to migrate was consistently weaker after 10 μM curcumin treatment compared to controls, *p* < 0.001 (Figure 3B). 

### 2.5. Curcumin Reduced NF-κB p65 Target Protein Survivin Expression and Led to an Accumulation of DNA Double-Strand Breaks in Curcumin-Sensitive CRC Cells

After incubation with 25 μM curcumin for 24 h, the immunoblot showed that the expression of the NF-κB p65 target protein survivin significantly decreased in HROC18, HROC24, and HROC357 cells, but not in HROC285 cells. The expression of Bcl-xL—another NF-κB p65 target protein—was unchanged in all cell lines except in curcumin-sensitive HROC18 cells, where it showed a lower expression. In addition, we also observed that curcumin induced accumulation of the DNA damage marker ɣH2AX in all four CRC cell lines—especially in curcumin-sensitive HROC357 cells (Figure 4 and Appendix A).

### 2.6. Curcumin Triggered Higher IκBα Levels in Curcumin-Sensitive CRC Cell Lines

The IκBα levels were measured after 0 µM (control), 10 μM, 15 μM, and 20 μM curcumin treatment for 24 h. In general, IκBα levels increased in all six CRC cell lines with rising curcumin concentrations (Figure 5A). Treatment with 20 μM curcumin triggered a 20–30% increase in total IκBα (*p* < 0.05, Figure 5A) in curcumin-resistant cell lines. However, for the curcumin-sensitive cell lines, a significant difference between control and curcumin treatment was already observable at 10 μM curcumin (*p* < 0.05). The increase in IκBα expression after exposure to 20 μM curcumin was as high as 70% (*p* < 0.05, Figure 5A).

### 2.7. Inverse Correlation of Cell Viability and IκBα Levels after Curcumin Treatment 

Next, correlations of cell viability and IκBα expression levels after curcumin treatment (10 μM, 15 μM, and 20 μM) were investigated. For curcumin treatment at 10 μM concentration, cell viability and IκBα levels in the CRC cell lines were strongly correlated (Figure 5B), and had a very high coefficient value (*r* = −0.943, *p* = 0.017). Similar inverse correlations were obtained for 15 μM (Figure 5B; *r* = −0.986 (*p* < 0.001)) and 20 μM curcumin (Figure 5B; *r* = −0.928 (*p* = 0.006)), confirming that the cell viability is negatively associated with the IκBα expression level in CRC cell lines after curcumin treatment.

### 2.8. Curcumin Downregulated NF-κB p65

We further measured the expression of NF-κB p65, as well as cytokines and chemokines induced by NF-κB p65 (Figure 6 and Appendix A). After incubation with 25 μM curcumin for 24 h, a decrease in NF-κB p65 expression was consistently seen in all four cell lines (HROC24, HROC285, HROC18, and HROC357)—especially in the sensitive cells HROC18 and HROC357—with statistically significant differences between controls and curcumin treatment (Figure 6A and Appendix A).

Moreover, no significant decrease in IL-6 secretion (Figure 6B) was observed after 7-day incubation with curcumin in these four cell lines. However, a significant decrease was found for IL-8 in HROC18 (Figure 6C) and for CXCL-1 in HROC18 and HROC357 cells (Figure 6D) under curcumin treatment.

### 2.9. Curcumin Induced Cell Death by Inhibiting NF-κB p65 Activation via IκBα

Based on the above results, we hypothesized that curcumin exhibits its anticancer effects via IκBα, which inhibits the NF-κB p65 nuclear translocation. In order to verify this, CRC cells were incubated with 50 µM caffeic acid phenethyl ester (CAPE)—an inhibitor of NF-κB—for 24 h. After that, an increase in IκBα levels was consistently seen in HROC18 and HROC24 cells (Figure 7A), and cell death levels subsequently significantly increased for both curcumin-resistant and -sensitive cells in comparison to controls (Figure 7B). Notably, the good-response to single treatment with CAPE was especially seen in curcumin-sensitive cell lines (Figure 7B).

Furthermore, we evaluated the IL-8 and CXCL-1 levels in CRC cells after incubation with curcumin and CAPE for 24 h. A significant decrease in IL-8 and CXCL-1 levels was consistently found in HROC18 and HROC24 cells compared to controls (Figure 7C,D). In HROC18 cells, even a single treatment with CAPE decreased IL-8 and CXCL-1 secretion (Figure 7C,D).

### 2.10. Mutated SMAD4 Was Correlated with Curcumin Sensitivity in CRC Cell Lines

Mutations in 10 common CRC genes—including *APC*, *K-Ras*, *B-Raf*, *PIK3CA*, *TP53*, *ATM*, *SMAD4*, *RAD50*, *POLE*, and *PTEN*—were investigated as possible biomarkers associated with the IC_50_ values of curcumin. CRC cell lines with mutated *SMAD4* were significantly more sensitive to curcumin (wild vs. mutant type, 12.32 ± 0.70 μM vs. 7.97 ± 0.49 μM, *p* = 0.024). No further correlations were found (Figure 8).

### 2.11. MSS CRC Cell Lines Were More Sensitive to Curcumin

Stratified by microsatellite (MS) status, the IC_50_ value of curcumin was significantly lower in microsatellite-stable (MSS) than in microsatellite-instable (MSI) CRC cell lines (MSS vs. MSI, 10.50 ± 0.66 μM vs. 13.67 ± 1.03 μM, *p* = 0.011) (Figure 9A). This supports the hypothesis that CRC cell lines of the MSS type are more sensitive to curcumin. When comparing the baseline IκBα expression levels in CRC cell lines with different MS status, we observed that the mean IκBα level in MSS CRC cell lines was significantly higher compared to MSI CRC cell lines (MSS vs. MSI, 25.37 ± 2.40 μM vs. 14.91 ± 2.72 μM, *p* = 0.008) (Figure 9B).

### 2.12. MSS Combined with High Baseline IκBα Levels Estimated Curcumin Sensitivity

Finally, the MS status and baseline IκBα expression levels were combined as biomarkers to estimate the sensitivity of CRC cell lines to curcumin. The area under the receiver operating characteristic (ROC) curve presented a statistically significant predictive value (area under the ROC curve (AUC) = 0.72, 95%CI: 0.54–0.90, *p* = 0.035), with a 72% probability to correctly distinguish curcumin-sensitive from curcumin-resistant CRC cell lines (Figure 9C). 

## 3. Discussion

The present study is the first report on the identification of a potential biomarker for curcumin sensitivity in CRC cells. Mutations in *APC*, *K-Ras*, *B-Raf*, *PIK3CA*, *TP53*, *ATM*, *SMAD4*, *RAD50*, *POLE*, and *PTEN* are usually considered to be relevant CRC biomarkers [29,30,31,32] as cancer therapy target structures in precision medicine approaches [33]. Here, we found only mutations of the gene *SMAD4* to be associated with curcumin sensitivity. Agarwal et al. [34] reported that curcumin suppressed the cell proliferation ability and induced apoptosis by upregulating reactive oxygen species in a colon cancer cell line with mutations in *SMAD4* and *TP53*; however, *SMAD4* was not considered as a biomarker with the ability to predict the sensitivity of CRC cells to curcumin. Even though 30 patient-derived low-passaged CRC cell lines with known *SMAD* mutational status—consisting of 4 mutated and 26 wild-type cell lines—were analyzed in the present study, this point needs to be further clarified in future studies using an even larger number of suited CRC cell models.

More, the MS status and *MMR* gene expression levels are regarded as important biomarkers in cancer progression, for prognosis of outcome, and even for assessment of the efficacy of adjuvant chemotherapy [35,36,37]. In our series, most MSI CRC cell lines were more resistant, whereas most MSS CRC cell lines were sensitive to curcumin.

Shakibaei et al. [38] reported that curcumin combined with 5-FU could affect MMR-deficient CRC cells; however, MMR-proficient CRC cells were more sensitive, thus matching our data on curcumin sensitivity. In another study of combinatorial treatment in vitro, Caco-2 cells (characterized by MSS) and DLD-1 cells (characterized by MSI) were treated with curcumin in combination with resveratrol and, again, the MSI cells were found to be more resistant [39]. The phenomenon of MSI CRC cell resistance to chemotherapeutic regimens has been ascribed to the so-called mutator phenotype as a result of MMR deficiency, which allows MSI cells to become tolerant to DNA-damaging agents [40,41,42]. 

Lynch syndrome (LS) patients, who are characterized by germline mutations in four of the *MMR* genes, have a high risk of cellular carcinogenesis caused by elevated oxidative stress [43,44,45]. However, nutritional Nrf2 activators (such as sulforaphane, curcumin, quercetin, and resveratrol) have been found to trigger the cellular antioxidative and inflammatory systems, thereby reducing disease progression risk factors in patients [46]. Since defective MMR underlies the MSI phenotype [47], it is reasonable to speculate that MSI CRC cells are adapted to high cellular oxidative stress, and that this might contribute to their higher resistance to curcumin compared to MMR-proficient CRC cells. 

However, several studies have come to differing conclusions. Majumdar et al. reported a better effect of curcumin together with resveratrol on MSI-positive HCT116 cells, but not on HT29 cells, which are MSS [48]. Wei et al. observed a potent apoptosis-inducing ability of curcumin on the MMR-deficient CRC cell lines HCT116, LoVo, SW48, and HCT15, but their study lacked proper MMR-proficient controls; thus, a higher sensitivity of MMR-deficient CRC cell lines cannot be concluded [49]. Jiang et al. suggested that the MMR system is strongly associated with the curcumin response in LS CRC [50]; in their study, an elevated cytotoxic response to curcumin was observed in MLH1-deficient HCT116 and RKO cells as well as in MSH2-deficient Hec59 cells compared to matching MMR-proficient HCT116+ch3, RKO+MLH1, and Hec59+ch2 cells [50]. However, since a limited sample number of classical CRC cell lines was used, off-target effects of the genetic modifications are possible and, most importantly, the MMR-proficient cell lines are still hypermutated, since the reintroduction of a defective MMR gene does not revert mutations already present in the cells; thus, a direct comparison to our data is not valid. 

In our analysis, curcumin could decrease cell viability and induce cell death in all 32 CRC cell lines tested. However, higher curcumin IC_50_ values were found in MSI compared to MSS CRC cell lines. In contrast to the contradictory studies mentioned above, all cell lines used in the present study were below passage number 50, and can thus still be considered as closely mirroring the biology of the original patients’ tumors [23].

Moreover, the key inhibitory protein of NF-κB p65 in the NF-κB signaling pathway—IκBα—was for the first time identified as an important biomarker with the potential to precisely predict curcumin sensitivity. We found that curcumin-sensitive cell lines expressed significantly higher levels of IκBα even without treatment, while curcumin-resistant cell lines frequently expressed low baseline IκBα levels. 

Mechanistically, the NF-κB signaling pathway represents the classical pathway responsible for the anticancer action of curcumin. Resistance to radiation therapy in most patients has been linked to an activation of the NF-κB pathway, and Sandur et al. suggested that inhibiting this pathway could resensitize tumors to radiotherapy [51]. Indeed, although radiation activated NF-κB, curcumin treatment could suppress the radiation-induced NF-κB pathway activation and prevent therapy resistance [51,52]. In detail, radiotherapy induced an elevation of NF-κB signaling by stimulating Akt phosphorylation, inhibitor of κB kinase (IKK) activation, and IκBα phosphorylation [51,52,53]. Similarly, downregulation of NF-κB signaling by curcumin—mainly through suppressing IKK activation and IκBα phosphorylation—could reverse 5-FU treatment resistance [54]. Curcumin enhanced the expression of proapoptotic proteins such as caspase-3, caspase-8, caspase-9, PARP, and Bax, and diminished the expression of antiapoptotic proteins such as Bcl-2 and Bcl-xL, as well as of cell proliferation proteins such as cyclin D1 [10,54,55]. 

Deregulation of NF-κB and its downstream targets has been proposed as an important targetable mechanism in CRC therapy resistance [56]. IκBα is considered to be a key protein for reversing the chemotherapy-induced upregulation of NF-κB signaling. This was explained by curcumin inhibiting the oxaliplatin-induced activation of NF-κB p65 and decreasing the expression of NF-κB antiapoptotic and pro-proliferative gene products [56]. 

Different gene expression patterns have been identified in oxaliplatin-resistant and -sensitive CRC cell lines after treatment with curcumin plus oxaliplatin. Compared to oxaliplatin-sensitive CRC cells, curcumin inhibited the NF-κB p65 level in oxaliplatin-resistant cells more effectively [57]. Similar results were also obtained in the present study, as we measured increasing IκBα levels in response to curcumin treatment in six CRC cell lines (three curcumin-sensitive and three curcumin-resistant ones). Compared to curcumin-resistant CRC cell lines, the curcumin-sensitive CRC cell lines expressed higher IκBα levels.

Taken together, experimental evidence has proven that curcumin treatment is associated with inhibition of NF-κB p65, activation of IκBα, suppression of IκBα phosphorylation, and downregulation of NF-κB-p65-related antiapoptotic and cell-proliferation-activating proteins [58,59]. It is thus conceivable that we could identify the IκBα baseline expression level as an important biomarker for prediction of curcumin sensitivity in CRC cell lines. 

Moreover, a significant correlation between the IC_50_ of curcumin and the baseline IκBα level in CRC cell lines was observed, along with a strong association of cell viability with the IκBα expression levels after curcumin treatment. Surprisingly, IκBα expression levels were higher in MSS compared to MSI CRC cell lines. When combining IκBα levels with MS status as a biomarker for predicting curcumin sensitivity, the combination could correctly distinguish the curcumin-sensitive CRC cell lines from curcumin-resistant CRC cell lines with an approximately 72% predictive power. The major finding of the present study—that MSS CRC cells have higher IκBα levels, and are more sensitive to curcumin treatment—is novel and surprising. Nevertheless, since these data were obtained with a relatively large number of low-passaged cell lines, even allowing for statistical comparison, they can be considered to be relevant. 

Our findings clearly raise the question of whether there is a link between the NF-κB pathway and MSI status. MSI cancers are prototypically hypermutated, and are well known to possess a variety of immune escape mechanisms to evade high immune pressure [60]. Among the first mechanisms identified was the modified antigen-processing machinery, in which the proteasome plays a central role [61]. Therefore, we suggest that this possibly interferes with the ability of curcumin to inhibit proteasomal degradation of IκBα. Additionally, a recent publication analyzed gene networks involved in oncogenic-associated inflammation, and observed that, among others, *MMR* gene expression was negatively correlated with inflammation-associated transcription factor (including NF-κB) expression [62]. Another finding in line with our results is the fact that the cellular cytotoxicity of temozolomide—a drug used for treatment of gliomas—has been at least partially attributed to its inhibitory effects on NF-κB [63]. More importantly, the ability of temozolomide to inhibit NF-κB activation was shown to be dependent on the presence of a functional MMR system [63]. 

In conclusion, data from different studies indicate that cancer responses to curcumin—especially CRC—are mediated by inhibition of NF-κB p65 via IκBα. This additionally seems to be linked to the MMR system. However, the exact mode of action still needs to be determined. 

## 4. Materials and Methods

### 4.1. CRC Cell Lines and Cell Culture

Directly patient-derived and patient-derived xenograft (PDX)-derived CRC cell lines (*n* = 32) were established and characterized as previously described [23], including information on cancer gene mutations and MS status [23,64,65]. Cells were cultured in DMEM/Ham’s F-12 medium containing 10% fetal calf serum (FCS) and 2 mM L-glutamine (all cell culture reagents from Pan-Biotech, Aiden, Germany), and grown in a humidified incubator (5% CO_2_ and 37 °C). Cells below passage 50 were used in order to ensure comparability of molecular characteristics between parental tumor and tumor-derived cell lines. 

### 4.2. Chemo-Test and Crystal Violet Assay

A total of 1.0–1.5 × 10^4^ CRC cells per well were seeded in a 96-well plate and allowed to adhere for 24 h. Cultures were treated with 1 μM–25 μM curcumin (Sigma-Aldrich, Darmstadt, Germany, CAS: 458-37-7), 1 nM–5000 µM 5-FU, 0.1 nM–1000 µM irinotecan, and 1.3 nM–20 µM oxaliplatin (pharmacy of the University Hospital Rostock, Rostock, Germany) [24], and incubated twice for 72 h each, with a medium change in between. 

Subsequently, cells were washed with PBS, then stained using 0.2% crystal violet solution (AppliChem, Darmstadt, Germany). After that, cells were again washed twice and allowed to dry completely. Photos were taken, and stained biomaterial present in the wells was quantified by dissolving in 1% SDS (AppliChem) and measuring the absorption at 570 nm (reference: 620 nm) [25,66]. Cell viability was calculated by the following formula: cell viability = (sample − dead control)/(living control − dead control) × 100%.

It is worth mentioning that we cited the results of IC_50_ values of standard drugs from our previous work. These values were obtained using the method described above. In detail, the cited IC_50_ values of 5-FU and irinotecan in HROC87 and HROC113 were from [27], in HROC57 from [28], and in HROC277 and HROC348 from [24]; the IC_50_ of irinotecan in HROC18 was from [26] and in HROC126 from [24]; while the IC_50_ of oxaliplatin in HROC57 was from [28], and in HROC277 and HROC348 from [24]. 

### 4.3. Cell Death Measurement

A total of 2 × 10^5^ CRC cells per well were seeded in a 6-well plate and left for incubation overnight. Cultures were exposed to curcumin and CAPE (Hycultec GmbH, Beutelsbach, Germany, cat: HY-N0274) as single or combination treatments for 72 h. Cell death was evaluated by apoptosis assay using Annexin V-FITC (ImmunoTools, Germany, cat: 31490013) and propidium iodide (PI, AppliChem, code: A2261) labeling and flow cytometric analysis on a FACSCalibur device (Becton Dickinson, East Rutherford, NJ, USA) with CellQuest™ Pro software (Becton Dickinson), and 20,000 events were recorded for each measurement. The data were analyzed using the following formula: 100% − (Annexin V^-^ PI^-^) % [66].

### 4.4. Evaluation of IκBα in Cells

In order to evaluate the baseline IκBα level, 5 × 10^5^ CRC cells were fixed using 400 μL of 2% Formafix (Formafix GmbH, Düsseldorf, Germany) followed by an incubation in 400 μL of 1 × buffer P (100× buffer P: 0.5 mL of FCS, 5 mL of 1% saponin, 5 mL of 0.1 M HEPES, and 39.5 mL of PBS) for 10 min. For staining, cells were incubated in 100 μL of 1 × buffer P with or without 5 μL of PE-anti- IκBα antibody (BioLegend, San Diego, CA, USA, cat: 662412), and protected from light for 30 min at 4 °C. IκBα expression was analyzed by flow cytometry as described above, and 10,000 events were recorded for each measurement.

To assess the IκBα levels in cells after treatment, cells were pretreated with 5% FCS medium containing curcumin and CAPE as single or combination treatments for 24 h, prior to PE-anti- IκBα staining.

### 4.5. Wound-Healing Assay

Fully confluent CRC cells were incubated in medium without FCS, with and without the addition of 10 μM curcumin, for 24 h. Afterwards, the monolayers were scratched using a 20 μL pipette tip to generate a wound in the cell layer. The wounds’ diameters were measured daily for up to 5 days using a light microscope (Microscope ZEISS Primovert, Oberkochen, Germany). 

### 4.6. Measurement of IL-6, IL-8, and CXCL-1 in Cells

A total of 5 × 10^5^ CRC cells were allowed to adhere in a 6-well plate overnight. Medium containing 5% FCS with or without 25 μM curcumin was changed, and cells were incubated for 7 days. Supernatants were collected and measured by ELISA according to the manufacturer’s recommendations for IL-6 (cat: 31670069) and IL-8 (cat: 31670089) (both from ImmunoTools, Friesoythe, Germany), as well as CXCL-1 (R&D Systems, Minneapolis, MN, USA, cat: DY275-05). In order to test the effects of CAPE on IL-8 and CXCL-1 levels, cells were incubated with curcumin, CAPE, or a combination of curcumin and CAPE for 24 h prior to measurement.

### 4.7. Immunoblots

A total of 5 × 10^5^ CRC cells per well were seeded in a 6-well plate. After incubation overnight, the medium was changed to 5% FCS medium plus/minus 25 μM curcumin. After 24 h, cells were pelleted and collected, and immunoblots were performed using anti-survivin (Abcam, Cambridge, UK, cat: ab134170), anti-ɣH2AX (Cell Signaling, Frankfurt/Main, Germany, cat: 9718), anti-Bcl-xL (Abcam, cat: ab32370), and anti-NF-κB p65 (Santa Cruz, Heidelberg, Germany, cat: sc-372). 

### 4.8. Statistical Analyses

Cell viability at gradient concentrations of curcumin was plotted by a heatmap visualization technique using the TBtools software [67]. Pearson’s or Spearman’s rank correlation analyses were performed according to the data’s distribution (normal or not) and presented as a correlation coefficient (*r*). Student’s *t*-test and ANOVA were used to compare the mean differences between groups, and linear regression was used for comparison of trends by the slopes of regression lines. All results were taken from at least three independent measurements. The predictive ability of biomarkers for curcumin sensitivity was analyzed using the ROC, and is presented as the AUC and the 95% confidence interval. *p*-Values lower than 0.05 were considered significant. All statistical analyses were performed using GraphPad Prism 6.0 (GraphPad Software, San Diego, CA, USA).

## 5. Conclusions

In summary, this study demonstrated that curcumin could inhibit CRC cell viability, increase cell apoptosis, and suppress cell migration. The 32 CRC cell lines analyzed varied greatly in their response to curcumin. The primary focus was to identify the subset of CRC cell lines that are highly sensitive to curcumin. By unraveling common features of the responsive cell lines, possible biomarkers were identified; these were subsequently suggested as predictors of curcumin response. Most curcumin-sensitive CRC cell lines had the following characteristics: MSS and/or high baseline IκBα expression levels. Meanwhile, curcumin-resistant CRC cell lines were mostly MSI, and showed low baseline IκBα levels. The predictive ability of this biomarker combination was statistically significant, with a predictive power of around 72% in a classical ROC/AUC analysis.

These data further fuel the ongoing efforts to improve the bioavailability of curcumin in order to pave the way for successful clinical application. Based on our large panel of patient-derived low-passaged CRC cell models with known sensitivity and biomarker expression, novel curcumin-derived candidate drugs could now easily be screened. Since the HROC model collection also includes a large number of matching in vivo PDX models, even the complete preclinical drug testing could be managed.

Based on the data obtained so far, we postulate that MSS patients with high baseline IκBα expression levels can benefit from clinical treatment with novel curcumin formulations and derivatives. Further studies will also have to analyze the combination of curcumin with standard chemotherapeutics.

## Figures and Tables

**Figure 1 cancers-14-01032-f001:**
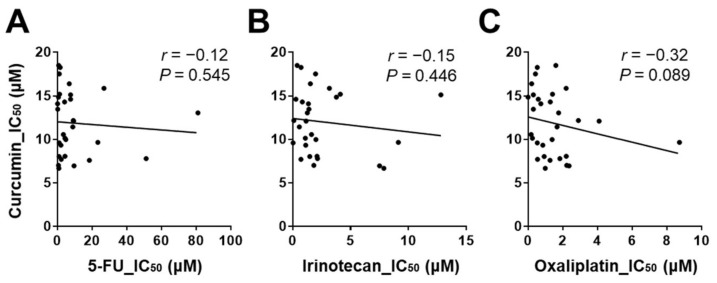
Sensitivity to curcumin plotted against sensitivity to standard drugs (5-FU, irinotecan, and oxaliplatin) in CRC cell lines: (**A**) Correlation analysis between the IC_50_ of curcumin and of 5-FU. (**B**) Correlation analysis between the IC_50_ of curcumin and of irinotecan. (**C**) Correlation analysis between the IC_50_ of curcumin and of oxaliplatin. These correlational analyses were performed using Spearman’s correlation analysis, and the correlation coefficient (*r*) was used to measure the strength of the relationship.

**Figure 2 cancers-14-01032-f002:**
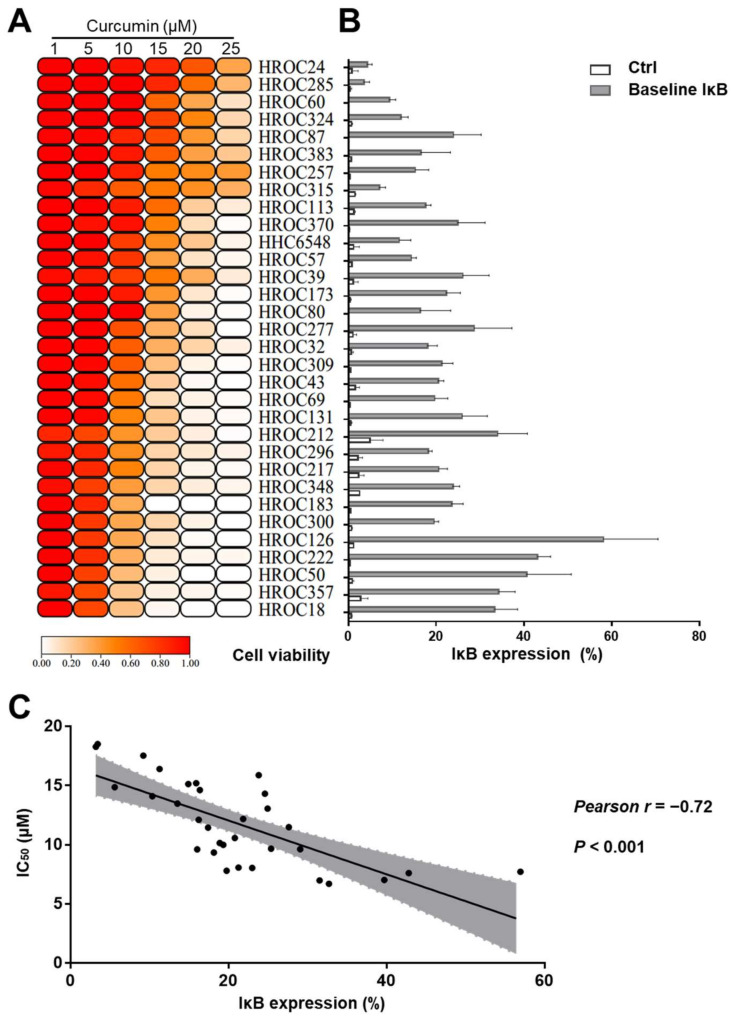
Association of curcumin sensitivity with baseline IκBα levels: (**A**) 32 CRC cell lines were tested for curcumin sensitivity via a long-term proliferation assay. The color scale, from red to white, indicates high to low cell viability, respectively. The heatmap was plotted with the help of TBtools software. (**B**) The baseline IκBα expression levels were measured in 32 CRC cell lines using flow cytometry. Control (Ctrl) group (white histograms) bars represent staining with an irrelevant control antibody, while the baseline IκBα group (grey histograms) bars represent staining with an IκBα antibody; the bars are depicted as the mean and SD. (**C**) The IC_50_ values of curcumin correlated with IκBα expression levels of CRC cell lines. This correlational analysis was performed using Pearson’s correlation analysis, and the correlation coefficient (*r*) was used to measure the strength of the relationship between curcumin’s IC_50_ and baseline IκBα expression levels in CRC cell lines.

**Figure 3 cancers-14-01032-f003:**
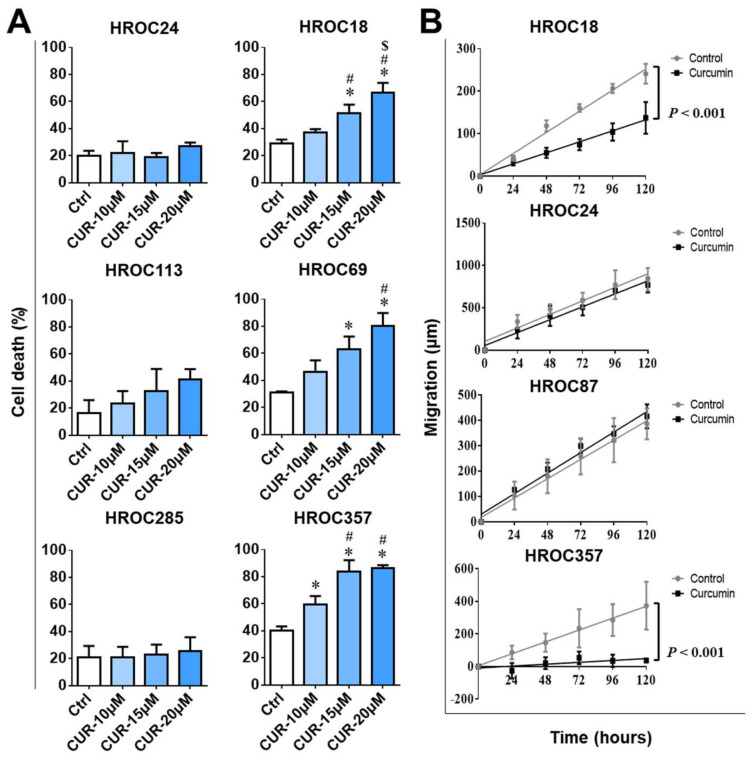
Curcumin induced cell death and suppressed cell migration in curcumin-sensitive CRC cell lines: (**A**) The cell death was measured by flow cytometry in three curcumin (CUR)-resistant (HROC24, HROC113, and HROC285) and three CUR-sensitive CRC cell lines (HROC18, HROC69, and HROC357) after treatment with 0 μM (Ctrl), 10 μM, 15 μM, and 20 μM CUR for 72 h. The statistical difference analysis was performed by one-way analysis of variance (ANOVA), and the bars represent the mean and SD—*: compared to control (Ctrl), #: compared to 10 μM CUR treatment, and $: compared to 15 μM CUR treatment; all of these markers showed *p* < 0.05. (**B**) Migration distance was assessed in fully confluent CUR-resistant (HROC24 and HROC87) and CUR-sensitive CRC cell lines (HROC18 and HROC357) after treatment with 10 μM CUR for 24 h. Subsequently, migration distance (μm) was determined by wound-healing assay. The statistical difference analysis was performed by linear regression, *p* < 0.001, compared to control (no-drug treatment).

**Figure 4 cancers-14-01032-f004:**
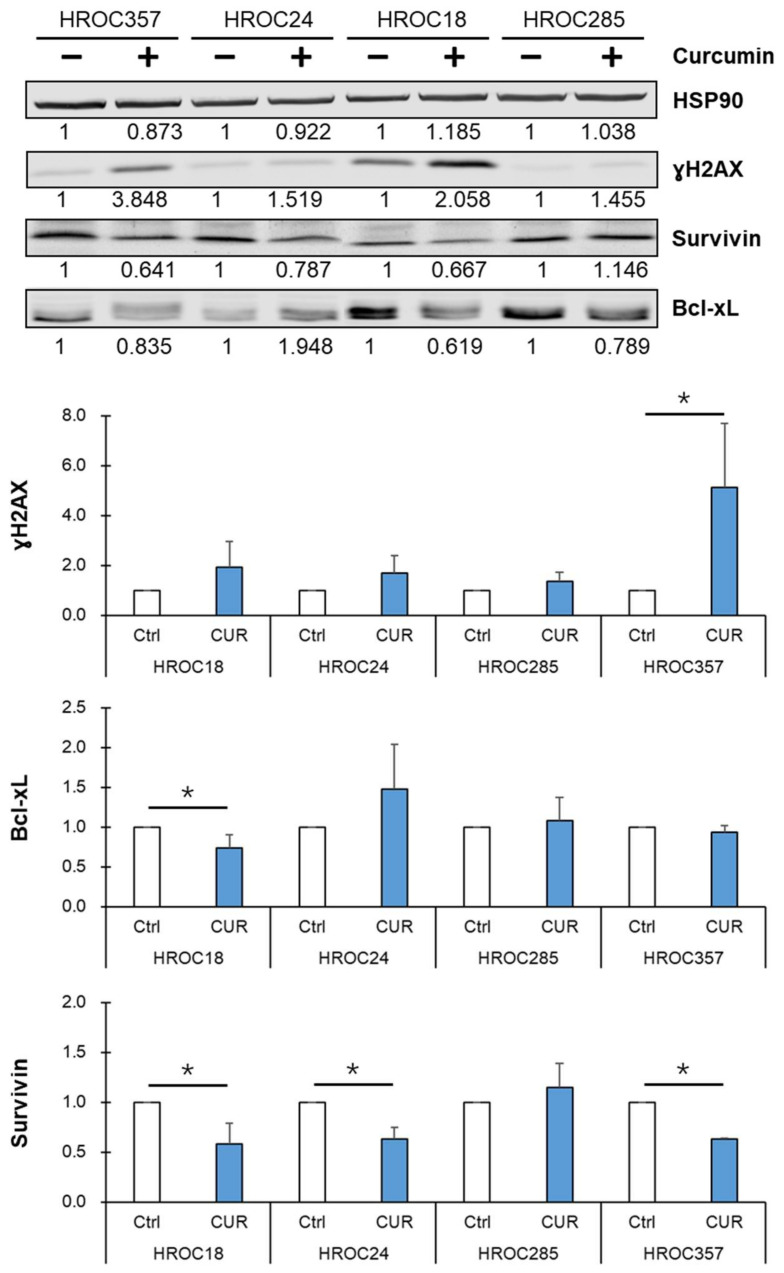
Curcumin reduced survivin protein expression and accumulation of DNA double-strand breaks in curcumin-sensitive CRC cells. Expression levels of ɣH2AX, survivin, and Bcl-xL were assessed via immunoblot in two CUR-sensitive (HROC18 and HROC357) and two CUR-resistant cell lines (HROC24 and HROC285) after pretreatment with 25 μM CUR for 24 h. The statistical difference analysis was performed using Student’s *t*-test, and bars represent mean and SD values—*: compared to controls (Ctrl, 0 μM CUR treatment), *p* < 0.05.

**Figure 5 cancers-14-01032-f005:**
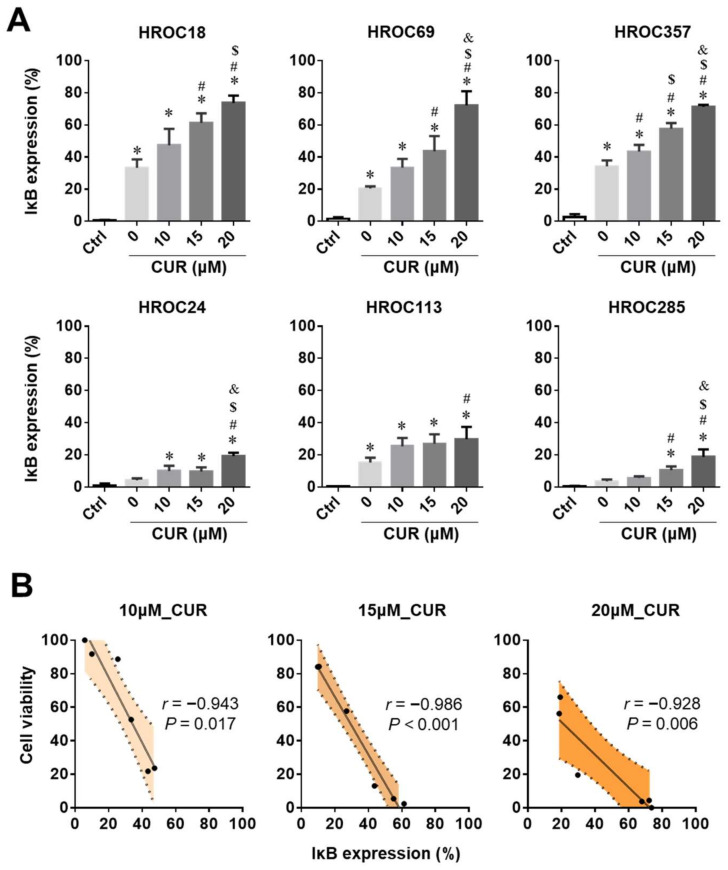
The cell viability is associated with the IκBα levels in CRC cells after curcumin treatment: (**A**) Assessment of IκBα expression levels by flow cytometry in three curcumin (CUR)-sensitive (HROC18, HROC69, and HROC357) and three CUR-resistant CRC cell lines (HROC24, HROC113, and HROC285) following treatment with 0 μM, 10 μM, 15 μM, and 20 μM CUR. The statistical difference analysis was performed by ANOVA, and the bars represent the mean and SD—*: compared to controls (Ctrl; staining with an irrelevant control antibody), #: compared to 0 μM CUR (staining with an IκBα antibody), $: compared to 10 μM CUR treatment, and &: compared to 15 μM CUR treatment; all of these markers indicate *p* < 0.05. (**B**) Correlation analysis of cell viability and IκBα levels after treatment with 10 μM, 15 μM, and 20 μM CUR. These correlational analyses were performed using Spearman’s correlation analysis, and represented by the correlation coefficient (*r*).

**Figure 6 cancers-14-01032-f006:**
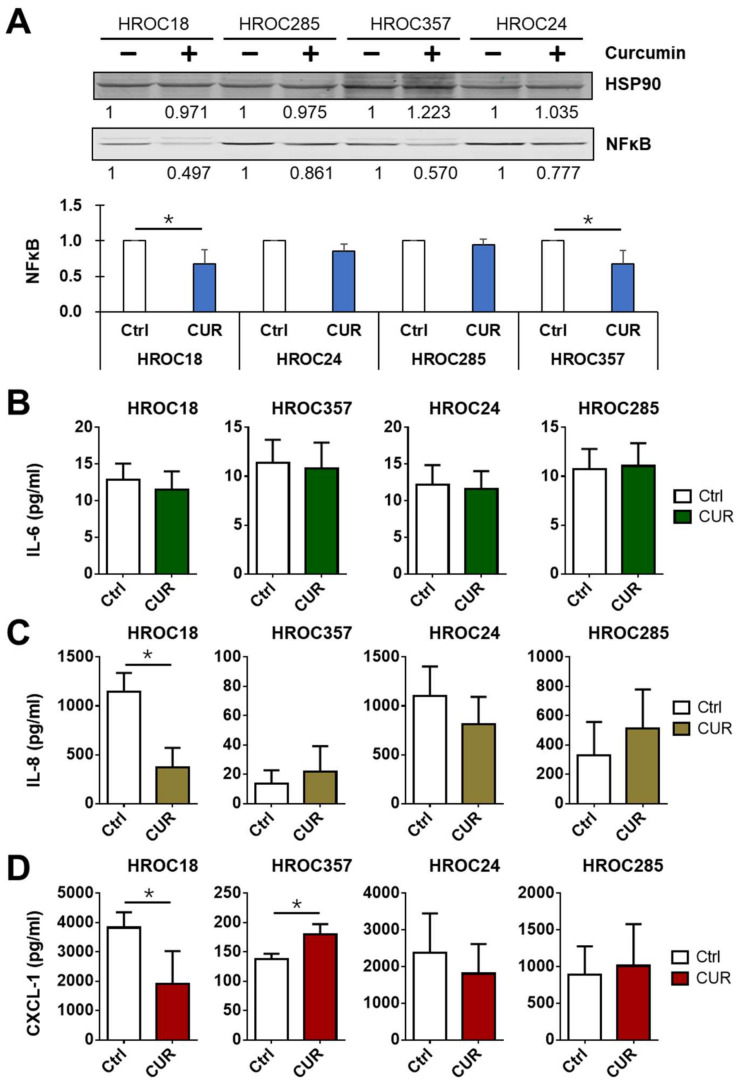
Effects of curcumin on NF-κB p65 and its targets: (**A**) The NF-κB p65 expression levels were measured in two curcumin (CUR)-sensitive (HROC18 and HROC357) and two CUR-resistant cell lines (HROC24 and HROC285) after pretreatment with 25 μM CUR for 24 h. NF-κB p65 was detected using immunoblots. (**B**) IL-6, (**C**) IL-8, and (**D**) CXCL-1 secretion levels were measured in two CUR-sensitive (HROC18 and HROC357) and two CUR-resistant cell lines (HROC24 and HROC285) after pretreatment with 25 μM CUR for 7 days. IL-6, IL-8, and CXCL-1 were detected using ELISA. The statistical difference analysis was performed using Student’s *t*-test, and bars represent mean and SD values—*: compared to control (Ctrl, 0 μM CUR treatment), *p* < 0.05.

**Figure 7 cancers-14-01032-f007:**
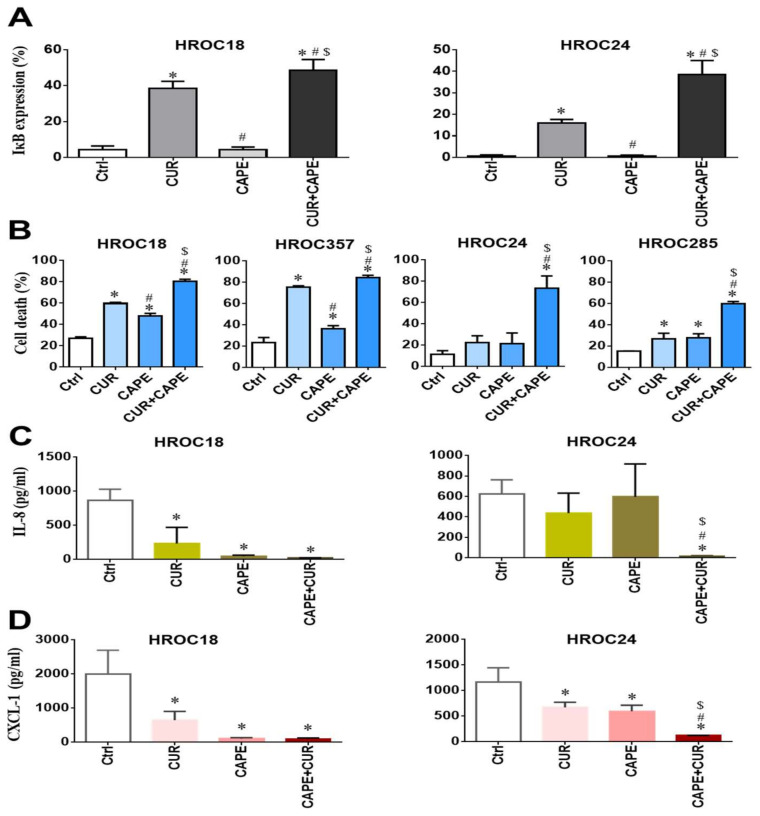
Effects of an NF-κB inhibitor (caffeic acid phenethyl ester (CAPE)) on CRC cells. After treatments for 24 h—control (no-drug treatment; Ctrl), 25 μM curcumin (CUR), 50 μM CAPE, and a combination of 25 μM CUR and 50 μM CAPE: (**A**) The IκBα expression levels were measured by flow cytometry in one CUR-sensitive (HROC18) and one CUR-resistant CRC cell line (HROC24). (**B**) Cell death was measured by flow cytometry in two CUR-sensitive (HROC18 and HROC357) and two CUR-resistant CRC cell lines (HROC24 and HROC285). (**C**) IL-8 and (**D**) CXCL-1 levels were measured by ELISA in HROC18 and HROC24 cell lines. All statistical difference analyses were performed by ANOVA, and the bars represent mean and SD values—*: compared to Ctrl, #: compared to 25 μM CUR treatment, and $: compared to 50 μM CAPE treatment; all of these markers indicate *p* < 0.05.

**Figure 8 cancers-14-01032-f008:**
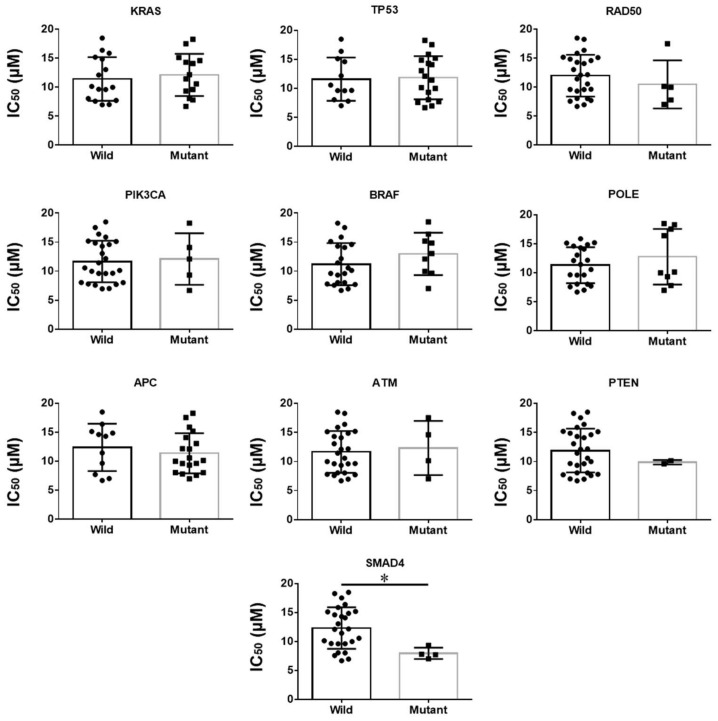
Comparison of curcumin IC_50_ (μM) values between wild and mutant types in 10 common CRC genes. The statistical difference analyses were performed using Student’s *t*-test, and the bars represent mean and SD values. *: *p* < 0.05.

**Figure 9 cancers-14-01032-f009:**
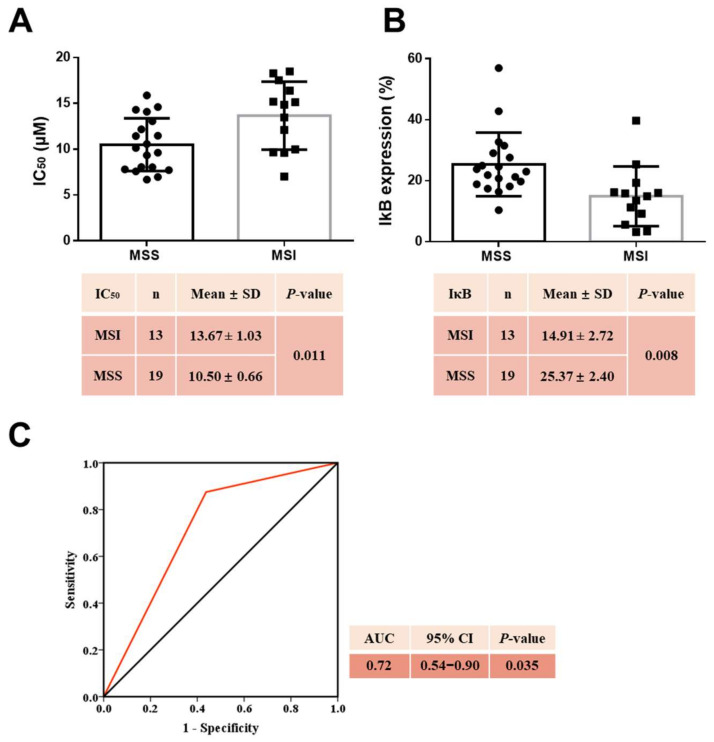
Prediction model for CRC cell line sensitivity to curcumin: (**A**) The comparison of IC_50_ (μM) values of curcumin between MSI and MSS CRC cell lines. Statistical difference analysis was performed using Student’s *t*-test, and the bars represent mean and SD values. (**B**) Comparison of the baseline IκBα levels in MSI and MSS CRC cell lines. Statistical difference analysis was performed using Student’s *t*-test, and the bars represent mean and SD values. (**C**) Predictive capacity of the baseline IκBα levels in CRC cells, and of MSI status, was evaluated by ROC and presented by the AUC. MSI: microsatellite instability; MSS: microsatellite stability; CI: confidence interval; solid circles indicate the IC_50_ values of MSS, and solid squares of MSI CRC cell lines.

**Table 1 cancers-14-01032-t001:** Patients and tumor characteristics.

Variables		*n*	%
Age (years) ^1^		32	66.34 ± 17.85
Gender	Male	15	46.88
	Female	17	53.13
Sample location	Colon	30	93.75
	Rectum	2	6.25
T stage	T0-T2	4	12.50
	T3-T4	28	87.50
N stage	N0	14	43.75
	N1-2	18	56.25
M stage	M0	22	68.75
	M1	9	28.13
	Mx	1	3.13
R stage	R0	23	71.88
	R1-2	8	25.00
	Unclear	1	3.13
L stage	L0	21	65.63
	L1	9	28.13
	Unclear	2	6.25
V stage	V0	19	59.38
	V1-2	11	34.38
	Unclear	2	6.25
UICC ^2^ stage	I-II	13	40.63
	III-IV	18	56.25
	Unclear	1	3.13
Molecular type	spStd ^3^	14	43.75
	spMSI-H ^4^	8	25.00
	CIMP-H ^5^	4	12.50
	Lynch	5	15.63
	Neuroendocrine	1	3.13

^1^ Age is presented as the mean ± SD; ^2^ UICC: Union for International Cancer Control stage; ^3^ SpStd: sporadic standard type; ^4^ spMSI-H: sporadic high-degree microsatellite instable type; ^5^ CIMP-H: highly CpG island methylated phenotype.

**Table 2 cancers-14-01032-t002:** IC_50_ values of curcumin and three standard drugs (5-FU, irinotecan, and oxaliplatin).

Cell Line	IC_50_ (μM)			
	Curcumin	5-FU	Irinotecan	Oxaliplatin
HROC24	18.49	0.79	0.39	1.60
HROC285	18.27	1.64	0.73	0.53
HROC324	17.52	1.17	1.98	0.43
HROC87	16.39	6.92 ^4^	1.53 ^4^	0.21
HROC60	15.87	26.90	3.17	2.19
HROC383	15.20	1.28	4.12	1.41
HROC113	15.12	7.69 ^4^	12.80 ^4^	0.31
HROC370	14.86	0.65	3.78	<0.01
HROC39	14.61	7.66	0.28	0.59
HROC80	14.30	4.21	0.83	1.28
HROC173	14.09	0.17	1.36	0.74
HHC6548	13.47	0.23	1.40	0.31
HROC57	13.05	80.95 ^5^	1.27 ^5^	1.76 ^5^
HROC277	12.17	9.20 ^1^	0.10 ^1^	2.90 ^1^
HROC257	12.11	9.35	1.16	4.09
HROC309	11.48	n.d.	n.d.	n.d.
HROC32	11.44	8.90	0.58	1.67
HROC43	10.57	3.51	1.63	0.18
HROC69	10.14	4.41	1.10	0.25
HROC131	9.99	4.89	2.01	1.39
HROC212	9.66	23.32	9.13	8.72
HROC296	9.62	n.d.	n.d.	n.d.
HROC315	9.60	1.08	0.04	0.53
HROC217	9.34	2.09	1.13	0.85
HROC348	8.06	4.40 ^1^	2.10 ^1^	2.20 ^1^
HROC183	8.04	1.05	1.48	0.93
HROC300	7.80	51.19	2.15	1.83
HROC126	7.72	2.30	0.72 ^2^	0.55
HROC222	7.60	18.47	n.d.	1.27
HROC50	7.02	0.66	1.83	2.23
HROC357	6.97	9.61	7.51	2.36
HROC18	6.69	0.88	7.88 ^3^	0.99
Mean ± SD	11.79 ± 3.55	9.83 ± 17.12	2.56 ± 3.03	1.48 ± 1.66
*p*-value ^6^		0.543	<0.001	<0.001

n.d.: Not determined; ^1^ IC_50_ has been reported previously: [24] ^2^, [25] ^3^, [26] ^4^, [27] ^5^, [28] ^6^, compared to the IC_50_ of curcumin.

## Data Availability

The data and materials are available from the corresponding author on reasonable request.

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
