# Peer review of "Microsatellite Status and IκBα Expression Levels Predict Sensitivity to Pharmaceutical Curcumin in Colorectal Cancer Cells"

_cancers, 2022, doi:10.3390/cancers14041032_

Round 1

Reviewer 1 Report

In this paper, authors highlight that ‘Microsatellite status and IκB-alpha expression levels predict sensitivity to pharmaceutical curcumin in colorectal cancer cells’ and claim that ‘a curcumins’ mode of action, inhibition of NF-κB p65 activation via IκB-alpha was identified’. 

However, I have several major concerns which need to be further explained. 

  1. Present results to support this statement: ‘After molecular classification, around 44% of the study population were sporadic standard (spStd), 25% sporadic high-degree microsatellite instable (spMSI-H), 16% Lynch-associated, 13% highly CpG island methylated (CIMP-H) as well as 3% neuroendocrine (Table 1).´ (Line 79-82). There is NO any data showing the microsatellite status. You confirmed it by yourself or it has been reported in another paper? What is the ‘molecular classification’?
  2. The molecular classification was done by testing the patient samples; however, all the work was done by using patient-derived xenograft (PDX)-derived CRC cell lines. You need to prove that all (PDX)-derived CRC cell lines still keep the same microsatellite (MS) status as the patient-derived.
  3. In order to prove this, authors checked IκB-alpha expression, several key factors (Such as IL-8 and CXCL- 1) and used the NF-κB inhibitor (caffeic acid phenethyl ester, CAPE), but never performed knockout or knockdown IκB-alpha work, a direct way to observe the correlation between curcumins action and IκB-alpha, which makes the conclusions less convincing.

Except the major concerns, several experiments and improvements are  required:

  1. Flow cytometry. Authors only provided the final calculated numbers without the Flow cytometry. It is hard to know the condition and the cell amount which were used during the experiment.
  2. IκB-alpha expression is an another key studied factor. In order to show that the flow based result is robust, Western blots or other methods to show the level of IκB-alpha protein expression, before and after the treatment.
  3. Figure 2B, photomicrograohs from microscopy are required.
  4. Western blots should have a statistical analysis in figure 2 and 5.
  5. Figure 3, cell cycle after 72hs? At certain high concentrations, most cells may already die.
  6. Sensitive and insensitive cells have different IC50, however, when studies were performed, much higher concentration, such as 25 uM was used, such as figure 5, 6 and 7. Why?
  7. Figure 9, IC50 MSI=13uM and IC50 MSS=10uM, even the data is significant, it is hard to draw the conclusion that MSS levels predicted curcumin-sensitivity.
  8. There are many fast and reliable products to measure the cell viability, why did you use crystal violet assay? Is there any special reason (method, Chemo-test and crystal violet assay)?

Author Response

Reviewer #1

Comments and Suggestions for Authors

In this paper, authors highlight that ‘Microsatellite status and IκB-alpha expression levels predict sensitivity to pharmaceutical curcumin in colorectal cancer cells’ and claim that ‘a curcumins’ mode of action, inhibition of NF-κB p65 activation via IκB-alpha was identified’.

However, I have several major concerns which need to be further explained.

  1. Present results to support this statement: ‘After molecular classification, around 44% of the study population were sporadic standard (spStd), 25% sporadic high-degree microsatellite instable (spMSI-H), 16% Lynch-associated, 13% highly CpG island methylated (CIMP-H) as well as 3% neuroendocrine (Table 1).´ (Line 79-82). There is NO any data showing the microsatellite status. You confirmed it by yourself or it has been reported in another paper? What is the ‘molecular classification’?

Author response:

We thank the reviewer for his/her attentive comment and sorry for this vague description. The molecular classification has been reported in previous studies, for example Mullins et al. (Ref: 23 in the manuscript). And we also presented the detailed data in the following Table 1 (only for reviewer #1).

[23] Mullins CS, Micheel B, Matschos S, Leuchter M, Bürtin F, Krohn M, Hühns M, Klar E, Prall F, Linnebacher M. Integrated Biobanking and Tumor Model Establishment of Human Colorectal Carcinoma Provides Excellent Tools for Preclinical Research. Cancers (Basel). 2019 Oct 9;11(10):1520. doi: 10.3390/cancers11101520. PMID: 31601052; PMCID: PMC6826890.

Table 1. Characteristics of tumors and cell lines

ID

-derived cell line

Molecular Subtype

Microsatellite status

Curcumin - IC50 (μM)

HROC24

Patient

spMSI-H

MSI

18.49

HROC285

PDX

Lynch

MSI

18.27

HROC324

Patient

Lynch

MSI

17.52

HROC87

PDX

spMSI-H

MSI

16.39

HROC60

Patient

CIMP-H

MSS

15.87

HROC383

Patient

spMSI-H

MSI

15.20

HROC113

Patient

Lynch

MSI

15.12

HROC370

Patient

spMSI-H

MSI

14.86

HROC39

Patient

spStd

MSS

14.61

HROC80

PDX

spStd

MSS

14.30

HROC173

Patient

spStd

MSS

14.09

HHC6548

PDX

Lynch

MSI

13.47

HROC57

Patient

neuroendocrine

MSS

13.05

HROC277

PDX

spStd

MSS

12.17

HROC257

Patient

spMSI-H

MSI

12.11

HROC309

Patient

spStd

MSS

11.48

HROC32

Patient

spStd

MSS

11.44

HROC43

Patient

CIMP-H

MSS

10.57

HROC69

Patient

spStd

MSS

10.14

HROC131

PDX

spMSI-H

MSI

9.99

HROC212

Patient

spMSI-H

MSI

9.66

HROC296

Patient

spStd

MSS

9.62

HROC315

PDX

Lynch

MSI

9.60

HROC217

PDX

spStd

MSS

9.34

HROC348

Patient

spStd

MSS

8.06

HROC183

PDX

CIMP-H

MSS

8.04

HROC300

PDX

CIMP-H

MSS

7.80

HROC126

Patient

spStd

MSS

7.72

HROC222

PDX

spStd

MSS

7.60

HROC50

PDX

spMSI-H

MSI

7.02

HROC357

Patient

spStd

MSS

6.97

HROC18

Patient

spStd

MSS

6.69

  1. The molecular classification was done by testing the patient samples; however, all the work was done by using patient-derived xenograft (PDX)-derived CRC cell lines. You need to prove that all (PDX)-derived CRC cell lines still keep the same microsatellite (MS) status as the patient-derived.

Author response:

Here, a misunderstanding of the models used must have taken place. In the Materials and Methods section, it was mentioned (Lines 436ff): “Directly patient-derived and patient-derived xenograft (PDX)-derived CRC cell lines (n = 32) were established and characterized as previously described, including information on cancer gene mutations and MS status [Ref: 23, 58, 59 of the revised version]”.

We used in the present study 32 CRC cell lines consisting of 20 patient-derived and 12 PDX-derived cell lines. Detailed information was given in Table 1 above (only for reviewer #1).

  1. In order to prove this, authors checked IκB-alpha expression, several key factors (Such as IL-8 and CXCL- 1) and used the NF-κB inhibitor (caffeic acid phenethyl ester, CAPE), but never performed knockout or knockdown IκB-alpha work, a direct way to observe the correlation between curcumins action and IκB-alpha, which makes the conclusions less convincing.

Author response:

We agree with this reviewer, that k.o. experiments (+ rescue) are the gold standard for functional proof. However, the major reason why we selected to use specific chemical inhibitors lies in the fact that the cell lines used were all in low passages – which is the closest to the original tumor biology as one can get in vitro and the reason why we like to work with patient-derived models instead of “old-fashioned” standard CRC cell lines. A major disadvantage of most of these low-passage cultures is their behavior of “difficult-to-be-transfected” and generally be genetically manipulated. For us, lack of convincing k.o. experiments are the minor evil compared to the chance to interrogate a high number of low-passaged patient-derived models for functional reactions. Thus, we would respectfully ask to accept this deficit in experimental design.

Except the major concerns, several experiments and improvements are required:

  1. Flow cytometry. Authors only provided the final calculated numbers without the Flow cytometry. It is hard to know the condition and the cell amount which were used during the experiment.

Author response:

We thank this reviewer for the helpful hint, and added the detailed conditions of Flow cytometry to the section of “Materials and Methods” in the improved version of the manuscript: 20,000 events were recorded for cell death measurement (Line 464); 10,000 events were detected for IκB-alpha measurement (Line 473). In addition, we also presented the original graphs in the following: Figure 1 for cell death measurement and Figure 2 for IκB-alpha measurement (both only for review #1).

Figure 1. Representative graphs for cell death measured by flow cytometry in three curcumin (CUR)-resistant (HROC24, HROC113, and HROC285) and three CUR-sensitive CRC cell lines (HROC18, HROC69, and HROC357) after treatment with 0μM (Ctrl), 10μM, 15μM, and 20μM CUR for 72 hours.

Figure 2. Representative graphs for IκB-alpha expression measured by flow cytometry in three curcumin (CUR)-resistant (HROC24, HROC113, and HROC285) and three CUR-sensitive CRC cell lines (HROC18, HROC69, and HROC357) after treatment with 0μM, 10μM, 15μM, and 20μM CUR for 24 hours. Control: cells stained with an isotype-matched irrelevant antibody.

  1. IκB-alpha expression is another key studied factor. In order to show that the flow based result is robust, Western blots or other methods to show the level of IκB-alpha protein expression, before and after the treatment.

Author response:

Another attentive suggestion from this reviewer. Here, we would like to refer to the product information data delivered by the supplier of the antibody clone used to stain for IκB-alpha in our study. The accurate suitability for IκB-alpha expression analysis of the antibody clone used has been carefully analyzed by the company Biolegend. Beside Flow cytometry, it is suitable and recommended for W-Blot analysis (see Figure 3; only for review #1). Thus, we would respectfully like to conclude that performing the analyses by Flow cytometry is sufficient and supports the conclusions drawn basing on this protein analysis method.

Clone: 3D6C02

Figure 3. Western blot of cell lysates from HeLa (Human), Raw264.7 (Mouse) and UMR106 (Rat) using IκB-α Mouse primary antibody (BioLegend, cat: 662412) and HRP Goat anti-Mouse secondary antibody (Cat. No. 405306). Direct-Blot™ HRP anti-β-actin (Cat. No. 643807) was used as a control.

  1. Figure 2B, photomicrographs from microscopy are required.

Author response:

The photomicrographs from microscopy were shown in the following Figure 4 (only for review #1).

Figure 4. The representative graphs for cell migration in two curcumin (CUR)-resistant (HROC24 and HROC87) and two CUR-sensitive CRC cell lines (HROC18 and HROC357) after treatment with 0μM (control) and 10μM CUR for 24 hours.

  1. Western blots should have a statistical analysis in figure 2 and 5.

Author response:

Similar to the point before, the statistical data was added and shown as Figure 4 and Figure 6A in the revised version of the manuscript.

Figure 4 in the revised version of the manuscript. Curcumin reduced NF-κB p65 target protein survivin expression and lead to an accumulation of DNA double strand breaks in curcumin-sensitive CRC cells. Expression levels of É£H2AX, Survivin and Bcl-xL were assessed using immunoblot in two CUR-sensitive (HROC18 and HROC357) and two CUR-resistant cell lines (HROC24 and HROC285) after pre-treatment with 25μM CUR for 24 hours. The statistical difference analysis was performed from three independent experiments by the student t-test, and bars represent mean and SD values; *, compared to control (Ctrl, 0μM CUR treatment), P < 0.05.

Figure 6 in the revised version of the manuscript. (A) The NF-κB p65 expression level was measured in two curcumin (CUR)-sensitive (HROC18 and HROC357) and two CUR-resistance cell lines (HROC24 and HROC285) after a pre-treatment with 25μM CUR for 24 hours. NF-κB p65 was detected using immunoblot. The statistical difference analysis was performed from three independent experiments by the student t-test, and bars represent mean and SD values; *, compared to control (Ctrl, 0μM CUR treatment), P < 0.05.

  1. Figure 3, cell cycle after 72hs? At certain high concentrations, most cells may already die.

Author response:

We want to thank this reviewer also for this very attentive comment. And indeed, the concentration of curcumin chosen in the measurement of cell cycle might have been inappropriate. After some internal discussions, we decided to remove the data of cell cycle measurement from the revised version of the manuscript. On the one hand it has been reported that curcumin affects CRC cells in vitro mainly by inducing a cell cycle arrest in the G2/M phase (Ref: 9). On the other hand, beside its lack of novelty, this finding was also not very relevant for our overall results and the conclusions drawn.

  1. Sensitive and insensitive cells have different IC50, however, when studies were performed, much higher concentration, such as 25 uM was used, such as figure 5, 6 and 7. Why?

Author response:

The reason behind that are the different incubation times. As described in the Materials and Methods section, we determine the IC50 by incubating twice for three days with the drugs to be analyzed; summing up to 6 days drug treatment. The data this attentive question aims at having been obtained from experiments using much shorter incubation times – which need higher drug concentrations to trigger measurable effects.

  1. Figure 9, IC50 MSI=13uM and IC50 MSS=10uM, even the data is significant, it is hard to draw the conclusion that MSS levels predicted curcumin-sensitivity.

Author response:

This comment is indeed hard to answer. What would this reviewer suggest as a consequence? Of course, we were looking for significant effects in our data – that was the reason why we could identify the MSS-status as potentially having a role in Curcumin’s action on CRC cells. Without doubt, this might trigger discussions and motivate others to check our observation using different CRC models. With all necessary respect: is this not the major idea of science?

  1. There are many fast and reliable products to measure the cell viability, why did you use crystal violet assay? Is there any special reason (method, Chemo-test and crystal violet assay)?

Author response:

Crystal violet was one the many methods we tested years ago side-by-side on the first approximately 10-15 patient-derived CRC cell cultures established in our lab. And it simply won that competition. Similarly, in that series of experiments, we also determined the twice 3 days incubation time for drug testing as being most reproducible and reliable. Part of that might be attributable to the doubling times of the CRC cultures used – they are in general longer than the old-fashioned standard lines.

Reviewer 2 Report

Review report for article under the title:

 Microsatellite status and IκB-alpha expression levels predict sensitivity to pharmaceutical curcumin in colorectal cancer cells

The manuscript has been aimed to identify new potential biomarkers for curcumin sensitivity as potential anticancer drug and its possible clinical application. The experimental study has been performed on thirty-two low-passage CRC cell lines. The authors come to the conclusion that curcumin suppressed cell proliferation in an individual manner related to CRC cell line. Among the curcumin-sensitive  CRC cell lines were those, microsatellite stable and expressed high levels of IκB. The authors hypothesized that novel curcumin formulations, either alone, but more likely in combination with standard therapeutics (???), can be expected to prove clinically beneficial for CRC cases with high IκB expression levels.

Main quality of  manuscript

  1. The methodology of reasearch is clearly presented in terms of the sample used, experimental  procedures and new omic measures; comprehensive ex vivo (individual) cell culture model may be very important and real new drug investigating platform. Since the concentrations of curcumin were a categorical factor, a complete blocking design was performed accurately.
  2. Statistical analyses are relevant in the context of the experiment, graphical data are clearly visible and can offer enough information, adding an additional value to the main subject of written paper.
  3. The manuscript is clearly written and understandable to read
  4. The focus of article is one approach toward precision or personalized medicine, aimed to determine an optimal therapy for each individual colorectal cancer cell culture sample.
  5. Deep molecular analysis of each tumor sample after dose-dependent curcumin treatment in regard to cell viability, NF-κB p65 target protein expression,  IκB-alpha level,  tumor stage,  É£H2AX, Survivin 150, Bcl-xL etc.

Major recommendation:

Having in mind that cell cultures are the basis for development of any new or complementary cancer treatment option,  it seems necessary to introduce one of the appropriate standard cancer treatment to make comparative analysis. New treatment should be compared to at least one of established treatments. In relation to individual cell culture, the question may be if new treatment is more effective, equally effective or less effective than established (standard) treatment(s)? If this is not done, then two independent manipulated variables are introduced (individual cell cultures and new treatment). This is because  the individual (original) colorectal cancer cell culture sample has been used with no comparison with standard colorectal cancer cells (like CaCo cells). It is necessary, especially when dependent variable(s) may represent also new markers. Many different drugs are approved to treat colorectal cancer. Since it is well documented that different treatment options and recommendations may depend on the type and stage of cancer, chemotherapy options, such as: Fluorouracil (5-FU), Irinotecan (Camptosar), Oxaliplatin (Eloxatin), Capecitabine (Xeloda) or Trifluridine/tipiracil (Lonsurf) or new targeted therapy options: Bevacizumab (Avastin), Regorafenib (Stivarga), Ziv-aflibercept (Zaltrap) and ramucirumab (Cyramza),  or EGFR inhibitors exist, at least one of them should be involved for comparison, according to the carcinoma stage.

Minor recommendations:

In driving a personalized approach, what may be the most important advances of modern oncology, the specific molecular alterations of each tumor and its specific response should be emphasized more, focusing on the results about the tumor heterogeneity which could be responsible for the lack of benefit when curcumin was used. For better explanation of therapeutical effectivity of a particular individual cancer type (personalized treatment), the excellent comprehensive article review of Gambardella et al (2020) should be considered.

Gambardella V, Tarazona N, Cejalvo JM, Lombardi P, Huerta M, Roselló S, Fleitas T, Roda D, Cervantes A. Personalized Medicine: Recent Progress in Cancer Therapy. Cancers (Basel). 2020 Apr 19;12(4):1009. doi: 10.3390/cancers12041009. PMID: 32325878; PMCID: PMC7226371.

Author Response

Reviewer #2

Comments and Suggestions for Authors

Review report for article under the title:

Microsatellite status and IκB-alpha expression levels predict sensitivity to pharmaceutical curcumin in colorectal cancer cells

The manuscript has been aimed to identify new potential biomarkers for curcumin sensitivity as potential anticancer drug and its possible clinical application. The experimental study has been performed on thirty-two low-passage CRC cell lines. The authors come to the conclusion that curcumin suppressed cell proliferation in an individual manner related to CRC cell line. Among the curcumin-sensitive CRC cell lines were those, microsatellite stable and expressed high levels of IκB. The authors hypothesized that novel curcumin formulations, either alone, but more likely in combination with standard therapeutics (???), can be expected to prove clinically beneficial for CRC cases with high IκB expression levels.

Main quality of manuscript

  1. The methodology of research is clearly presented in terms of the sample used, experimental procedures and new omic measures; comprehensive ex vivo (individual) cell culture model may be very important and real new drug investigating platform. Since the concentrations of curcumin were a categorical factor, a complete blocking design was performed accurately.
  2. Statistical analyses are relevant in the context of the experiment, graphical data are clearly visible and can offer enough information, adding an additional value to the main subject of written paper.
  3. The manuscript is clearly written and understandable to read
  4. The focus of article is one approach toward precision or personalized medicine, aimed to determine an optimal therapy for each individual colorectal cancer cell culture sample.
  5. Deep molecular analysis of each tumor sample after dose-dependent curcumin treatment in regard to cell viability, NF-κB p65 target protein expression, IκB-alpha level, tumor stage, É£H2AX, Survivin 150, Bcl-xL etc.

Author response:

We thank this reviewer for the very encouraging rating of our work.

Major recommendation:

Having in mind that cell cultures are the basis for development of any new or complementary cancer treatment option, it seems necessary to introduce one of the appropriate standard cancer treatment to make comparative analysis. New treatment should be compared to at least one of established treatments. In relation to individual cell culture, the question may be if new treatment is more effective, equally effective or less effective than established (standard) treatment(s)? If this is not done, then two independent manipulated variables are introduced (individual cell cultures and new treatment). This is because the individual (original) colorectal cancer cell culture sample has been used with no comparison with standard colorectal cancer cells (like CaCo cells). It is necessary, especially when dependent variable(s) may represent also new markers. Many different drugs are approved to treat colorectal cancer. Since it is well documented that different treatment options and recommendations may depend on the type and stage of cancer, chemotherapy options, such as: Fluorouracil (5-FU), Irinotecan (Camptosar), Oxaliplatin (Eloxatin), Capecitabine (Xeloda) or Trifluridine/tipiracil (Lonsurf) or new targeted therapy options: Bevacizumab (Avastin), Regorafenib (Stivarga), Ziv-aflibercept (Zaltrap) and ramucirumab (Cyramza),  or EGFR inhibitors exist, at least one of them should be involved for comparison, according to the carcinoma stage.

Author response:

We thank this reviewer for pointing this out, and we agree for the most part. The effects of the standard drugs in clinical routine for CRC (i.e. 5-FU, oxaliplatin and irinotecan) are tested routinely in our lab for novel cell cultures established – however, with some time lag. Since such a comparison was not intended, when the experiments for the present manuscript were performed, we have no complete data, but almost. For two cell lines used there are no reliable data available yet – and for one cell line, the irinotecan response is missing. The data are summarized and presented in the following Table, which was added as result as Table 2 and Figure 1 into the revised version of the manuscript.

When correlating the IC50 values, it became obvious that there might be a trend of negative correlation between response to curcumin and response to oxaliplatin. However, since significance was not reached, we did not discuss this potentially interesting side observation.

Table 2. IC50 values of curcumin and three standard drugs (5-FU, irinotecan and oxaliplatin)

Cell line

IC50 (μM)

Curcumin

5-FU

Irinotecan

Oxaliplatin

HROC24

18.49

0.79

0.39

1.60

HROC285

18.27

1.64

0.73

0.53

HROC324

17.52

1.17

1.98

0.43

HROC87

16.39

6.92d

1.53d

0.21

HROC60

15.87

26.90

         3.17

2.19

HROC383

15.20

1.28

4.12

1.41

HROC113

15.12

 7.69d

 12.80d

0.31

HROC370

14.86

0.65

3.78

< 0.01

HROC39

14.61

7.66

0.28

0.59

HROC80

14.30

4.21

0.83

1.28

HROC173

14.09

0.17

1.36

0.74

HHC6548

13.47

0.23

1.40

 0.31

HROC57

13.05

80.95e

1.27e

1.76e

HROC277

12.17

 9.20a

0.10a

2.90a

HROC257

12.11

9.35

1.16

4.09

HROC309

11.48

n.d.

n.d.

n.d.

HROC32

11.44

8.90

0.58

1.67

HROC43

10.57

3.51

1.63

0.18

HROC69

10.14

4.41

 1.10

0.25

HROC131

9.99

4.89

 2.01

1.39

HROC212

9.66

23.32

9.13

8.72

HROC296

9.62

n.d.

n.d.

n.d.

HROC315

9.60

1.08

0.04

0.53

HROC217

9.34

2.09

1.13

0.85

HROC348

8.06

4.40a

2.10a

2.20a

HROC183

8.04

1.05

1.48

0.93

HROC300

7.80

51.19

2.15

1.83

HROC126

7.72

2.30

0.72b

0.55

HROC222

7.60

18.47

n.d.

1.27

HROC50

7.02

0.66

1.83

2.23

HROC357

6.97

9.61

7.51

2.36

HROC18

6.69

0.88

7.88c

0.99

n.d.: not determined; a. IC50 was collected from Ref: 24; b. from Ref: 25; c, from Ref: 26; d, from Ref: 27; e, from Ref: 28 in the version of revised manuscript.

Figure 1. Sensitivity to curcumin plotted against sensitivity to standard drugs (5-FU, irinotecan and oxaliplatin) in CRC cell lines. (A) Correlation analysis between IC50 of curcumin and of 5-FU. (B) Correlation analysis between IC50 of curcumin and of Irinotecan. (C) Correlation analysis between IC50 of curcumin and of Oxaliplatin. These correlation analyses were performed by Spearman analysis, and the correlation coefficient (r) was used to measure the strength of the relationship.

Minor recommendations:

In driving a personalized approach, what may be the most important advances of modern oncology, the specific molecular alterations of each tumor and its specific response should be emphasized more, focusing on the results about the tumor heterogeneity which could be responsible for the lack of benefit when curcumin was used. For better explanation of therapeutical effectivity of a particular individual cancer type (personalized treatment), the excellent comprehensive article review of Gambardella et al (2020) should be considered.

Gambardella V, Tarazona N, Cejalvo JM, Lombardi P, Huerta M, Roselló S, Fleitas T, Roda D, Cervantes A. Personalized Medicine: Recent Progress in Cancer Therapy. Cancers (Basel). 2020 Apr 19;12(4):1009. doi: 10.3390/cancers12041009. PMID: 32325878; PMCID: PMC7226371.

Author response:

We want to thank the reviewer for this hint – and consistently added the work of Gambardella V et al. as Ref: 33 into the revised version of the manuscript (Line 319).

Round 2

Reviewer 1 Report

It is obvious that the authors made a great effort to revise the manuscript and have explained most of the questions. However, there are still several minor comments which I believe the authors need to fix. 

  1. I have read the reference of 58 – A novel curcumin analog inhibits canonical and non-canonical functions of telomerase through STAT3 and NF-κB inactivation in colorectal cancer cells. However, I do not think it has included information on cancer gene mutations and MS status to support your study.
  2. Authors explained that – We agree with this reviewer, that k.o. experiments (+ rescue) are the gold standard for functional proof. However, the major reason why we selected to use specific chemical inhibitors lies in the fact that the cell lines used were all in low passages – which is the closest to the original tumor biology as one can get in vitro and the reason why we like to work with patient-derived models instead of “old-fashioned” standard CRC cell lines. A major disadvantage of most of these low-passage cultures is their behavior of “difficult-to-be-transfected” and generally be genetically manipulated. For us, lack of convincing k.o. experiments are the minor evil compared to the chance to interrogate a high number of low-passaged patient-derived models for functional reactions. Thus, we would respectfully ask to accept this deficit in experimental design.                                                                                              Transient K.D experiments like siRNA only takes 48-72hs, almost the same time as you do the cell viability assay.  Moreover, if the low-passage cultures cells “difficult-to-be-transfected”, authors definitely can use cell lines to prove the correlation between curcumins action and IκB-alpha. However, consider the condition of using patient-derived samples, it is ONLY a suggestion.
  3. Sensitive and insensitive cells have different IC50, however, when studies were performed, much higher concentration, such as 25 uM was used, such as figure 5, 6 and 7. Why?                                     Author response:                                                                                  The reason behind that are the different incubation times. As described in the Materials and Methods section, we determine the IC50 by incubating twice for three days with the drugs to be analyzed; summing up to 6 days drug treatment. The data this attentive question aims at having been obtained from experiments using much shorter incubation times – which need higher drug concentrations to trigger measurable effects.                                          I cannot agree with this response. In Figure 6A, author used 25uM for 24hours to ’stimulate’ the effects, the IC50 of HROC357 and HROC18 is around 7um and the IC50 of HROC24 and HROC285 is around 20 uM, the used concentration 25uM is more close to HROC24 and HROC285 but almost 4 times of the sensitive ones. Then the conclusion – ‘especially in the sensitive cells HROC18 and HROC357 with statistically significant differences between control and 25μM curcumin treatment’ is weak. I suggest author to use 10um for sensitive and 25uM for resistant to repeat this experiment, if you still see a similar effect then this conclusion of ’statistically significant differences’ is solid.   
  1. Figure 9, IC50 MSI=13uM and IC50 MSS=10uM, even the data is significant, it is hard to draw the conclusion that MSS levels predicted curcumin-sensitivity.                                                                         Author response:                                                                                            This comment is indeed hard to answer. What would this reviewer  suggest as a consequence? Of course, we were looking for significant effects in our data – that was the reason why we could identify the MSS-status as potentially having a role in Curcumin’s action on CRC cells. Without doubt, this might trigger discussions and motivate others to check our observation using different CRC models. With all necessary respect: is this not the major idea of science?                                                                                                         I agree with authors, however, there is a doubt to define 10uM is ‘sensitive’ than 13uM, then it seems too ambitious to say ‘predict’. I suggest author to change the subtitle of ’MSS combined with high baseline IκB-alpha levels predicted curcumin-sensitivity’.

Author Response

Reviewer #1:

Comments and Suggestions for Authors: It is obvious that the authors made a great effort to revise the manuscript and have explained most of the questions. However, there are still several minor comments which I believe the authors need to fix.

  1. I have read the reference of 58 – A novel curcumin analog inhibits canonical and non-canonical functions of telomerase through STAT3 and NF-κB inactivation in colorectal cancer cells. However, I do not think it has included information on cancer gene mutations and MS status to support your study.

Author response: We are sorry for this mistake, in fact, we cited the reference 58 only for the following sentence “Taken together, experimental evidence has proven that curcumin treatment is associated with inhibition of NF-κB p65, activation of IκB-alpha, suppression of IκB-alpha phosphorylation as well as down-regulation of NF-κB p65 related anti-apoptotic and cell proliferation activating proteins [58, 59]” in the revised version of the manuscript.

And we cited references 23, 64 and 65 for the sentence this reviewer mentioned: “Directly patient-derived and patient-derived xenograft (PDX)-derived CRC cell lines (n = 32) were established and characterized as previously described [23], including information on cancer gene mutations and MS status [23, 64, 65]” in the revised version of the manuscript.

  1. Authors explained that – We agree with this reviewer, that k.o. experiments (+ rescue) are the gold standard for functional proof. However, the major reason why we selected to use specific chemical inhibitors lies in the fact that the cell lines used were all in low passages – which is the closest to the original tumor biology as one can get in vitro and the reason why we like to work with patient-derived models instead of “old-fashioned” standard CRC cell lines. A major disadvantage of most of these low-passage cultures is their behavior of “difficult-to-be-transfected” and generally be genetically manipulated. For us, lack of convincing k.o. experiments are the minor evil compared to the chance to interrogate a high number of low-passaged patient-derived models for functional reactions. Thus, we would respectfully ask to accept this deficit in experimental design.

Transient K.D experiments like siRNA only takes 48-72hs, almost the same time as you do the cell viability assay. Moreover, if the low-passage cultures cells “difficult-to-be-transfected”, authors definitely can use cell lines to prove the correlation between curcumins action and IκB-alpha. However, consider the condition of using patient-derived samples, it is ONLY a suggestion.

Author response: We thank the reviewer for this suggestion which we consider to follow in subsequent studies.

  1. Sensitive and insensitive cells have different IC50, however, when studies were performed, much higher concentration, such as 25 uM was used, such as figure 5, 6 and 7. Why?

 Author response: The reason behind that are the different incubation times. As described in the Materials and Methods section, we determine the IC50 by incubating twice for three days with the drugs to be analyzed; summing up to 6 days drug treatment. The data this attentive question aims at having been obtained from experiments using much shorter incubation times – which need higher drug concentrations to trigger measurable effects.

I cannot agree with this response. In Figure 6A, author used 25uM for 24hours to ’stimulate’ the effects, the IC50 of HROC357 and HROC18 is around 7um and the IC50 of HROC24 and HROC285 is around 20 uM, the used concentration 25uM is more close to HROC24 and HROC285 but almost 4 times of the sensitive ones. Then the conclusion – ‘especially in the sensitive cells HROC18 and HROC357 with statistically significant differences between control and 25μM curcumin treatment’ is weak. I suggest author to use 10um for sensitive and 25uM for resistant to repeat this experiment, if you still see a similar effect then this conclusion of ’statistically significant differences’ is solid.  

Author response:

In addition to the argumentation of the differences in incubation time, we want to add the following: Figure 2 shows that IkB levels correlate with curcumin sensitivity. We further demonstrate that curcumin modulates NF-κB signatures (Figs. 4-7). However, there is no clear association between NF-κB target gene modulation by curcumin or IkB modulation by curcumin and curcumin sensitivity (see e.g., Fig. 6A). This shows that the basal NF-κB activity - or another process that is dictated by IkB - are critical for sensitivity. Therefore, additional experiments with lower doses of curcumin will not be informative. To accomplish the reviewer comments, we now phrase more carefully and clearly in the revised version of the manuscript.

Indeed, a lack of correlation between NF-κB induction and curcumin sensitivity is not unexpected, given that several initial papers which claimed such a correlation were recently retracted; see e.g., https://www.surgjournal.com/article/S0039-6060(11)00696-9/pdf

https://journals.plos.org/plosone/article?id=10.1371/journal.pone.0222130

  1. Figure 9, IC50 MSI=13uM and IC50 MSS=10uM, even the data is significant, it is hard to draw the conclusion that MSS levels predicted curcumin-sensitivity.

Author response: This comment is indeed hard to answer. What would this reviewer suggest as a consequence? Of course, we were looking for significant effects in our data – that was the reason why we could identify the MS-status as potentially having a role in Curcumin’s action on CRC cells. Without doubt, this might trigger discussions and motivate others to check our observation using different CRC models. With all necessary respect: is this not the major idea of science?

I agree with authors, however, there is a doubt to define 10uM is ‘sensitive’ than 13uM, then it seems too ambitious to say ‘predict’. I suggest author to change the subtitle of ’MSS combined with high baseline IκB-alpha levels predicted curcumin-sensitivity’.

Author response: We thank the reviewer for his/her rigorous suggestion, and this subtitle now reads “MSS combined with high baseline IκB-alpha levels estimated curcumin-sensitivity” in the revised version of the manuscript.

Reviewer 2 Report

1) If the results shown in Table 2 (comparative analysis of IC50) are comparative data in the literature (citing references 24-26), then given table make a confusion what are your own results and what the results from the literature. In other words, there is a confusion as to whether the article represents an original work or a review based on the literature data. In order to avoid this confusion, which may at first sight be considered as a plagiarism of others authors results, my suggestion would be to point out significant difference in regard to IC50 of standard drugs and cite appropriate (corresponding to cell culture) literature data. 2) As concerns the results given in Figure 1 it is not clear whether the results are borrowed from literature or yours. In any other form, this table only creates confusion which cell type you have observed in your experiment and how the data obtained in different conditions may be compared may be used as yours. The same problem appears for Figure 1. You should rewrite the section Materials and methods to avoid this misunderstanding and to explain the experiment in detail, shown in Figure 1.

Author Response

Reviewer #2:

Comments and Suggestions for Authors

1) If the results shown in Table 2 (comparative analysis of IC50) are comparative data in the literature (citing references 24-26), then given table make a confusion what are your own results and what the results from the literature. In other words, there is a confusion as to whether the article represents an original work or a review based on the literature data. In order to avoid this confusion, which may at first sight be considered as a plagiarism of others authors results, my suggestion would be to point out significant difference in regard to IC50 of standard drugs and cite appropriate (corresponding to cell culture) literature data.

Author response: We thank this reviewer for the attentive comment, and added the statistically difference (P-values) between the IC50 values of standard drugs and curcumin in the revised version of the manuscript (Table 2 and line 114ff).

2) As concerns the results given in Figure 1 it is not clear whether the results are borrowed from literature or yours. In any other form, this table only creates confusion which cell type you have observed in your experiment and how the data obtained in different conditions may be compared may be used as yours. The same problem appears for Figure 1. You should rewrite the section Materials and methods to avoid this misunderstanding and to explain the experiment in detail, shown in Figure 1.

Author response: Similar to the point above, we agree with this reviewer. It is worth mentioning that the references we cited are all from our group, and we used the same methods of cell culturing.

In order to avoid misunderstanding, we have added the following paragraph to the Materials and Methods section, which reads “It is worth mentioning that we cited the results of IC50 values of standard drugs from our previous work. These values were obtained using the method as describe above. In detail, the IC50 values of 5-FU and irinotecan in HROC87 and HROC113 cited were from [27], in HROC57 from [28], in HROC277 and HROC348 from [24]; the IC50 of irinotecan in HROC18 from [26] and in HROC126 from [24]; the ones of oxaliplatin in HROC57 from [28], HROC277 and HROC348 from [24].” in line 506ff of the revised version of the manuscript.